# The Russo-Ukrainian War's toll on paediatric health during the first two years and future research directions: a scoping review
Filippa Sennersten[1], Safora Frogh[2], Sebastian Påhlsson[2], Andreas Wladis [2], Aida Alvinius[3] & Denise Bäckström[2,3] ✉

**Background** The Russo-Ukrainian War has profoundly affected healthcare systems, with children among the most vulnerable. Disruptions to essential services and care provision pose serious risks to child health and well-being. Understanding these impacts is critical to developing effective interventions and guiding research and policy. This scoping review examines paediatric health impacts during the Russo-Ukrainian War's first two years and identifies gaps in the evidence base.

**Methods** A PubMed search was conducted and supplemented with grey literature from WHO and UNICEF. Articles were included if they addressed children and focused on health-related impacts of the war. Eligible studies were required to be in English and published after the full-scale invasion on February 24, 2022. Exclusion criteria included articles unrelated to children, non-health topics, or geographically irrelevant studies. Data were synthesised thematically across key areas, including healthcare access, mental health, and chronic diseases. Risk of bias was assessed qualitatively.

**Results** Of 559 articles retrieved, 102 met the inclusion criteria. These included original research, reviews, letters, commentaries, and other relevant publications. The review found widespread disruptions in access to essential services and healthcare. It also identified significant impacts on injury and trauma care, chronic conditions, and infectious diseases. Challenges for displaced children and host countries were also frequently reported.

**Conclusions** Findings highlight the war's broad and complex impact on paediatric health and underscore the need for targeted responses. Addressing critical gaps in empirical research is essential to inform effective health policies and interventions.

In 2022, 468 million children globally lived in areas affected by armed conflict, with nearly 200 million in the world's most lethal war zones: the highest number in over a decade[1]. This alarming increase is in part driven by the ongoing war in Ukraine, which has placed children in conflict zones at severe risk. According to the World Health Organisation (WHO), health is defined as "a state of complete physical, mental, and social well-being, and not merely the absence of disease or infirmity"[2]. This broad definition underscores the widespread impact of conflict, affecting not only physical and mental health but also overall well-being. Children in conflict zones endure significant threats, including disrupted education, exposure to violence, and violations of their rights[1,3]. The increasingly urban nature of modern conflicts exacerbates these dangers, leaving children exceedingly

[1]Karolinska Institutet, Stockholm, Sweden. [2]Department of Biomedical and Clinical Sciences, Linköping University, Linköping, Sweden. [3]Department of Leadership and Command & Control, Swedish Defence University, Karlstad, Sweden. ✉e-mail: denise.backstrom@liu.se

vulnerable to long-term consequences of war[4–7]. This scoping review aims to provide an examination of how the conflict in Ukraine has affected paediatric health.

Before the conflict, Ukraine's healthcare system was relatively well-established, though it faced challenges[8,9]. The system was marked by a combination of Soviet-era infrastructure and ongoing reforms aimed at aligning with European standards. Healthcare for children was provided through a dual system, where approximately 50% of children were served by paediatricians and the other 50% by family physicians[10]. Additionally, a network of both specialised and non-specialised paediatric hospitals, as well as polyclinics, was involved in care provision. However, the system struggled with issues such as outdated medical equipment, regional disparities in healthcare access, and a reliance on institutional care for children with disabilities[9,11]. Vaccination coverage was low compared to other European countries, and the system faced challenges in managing chronic diseases, particularly in rural areas[8]. The mental healthcare system was dominated by psychiatric institutions with limited community services, and suffered from strong stigma, inadequate funding, and corruption[12]. The conflict has exacerbated these pre-existing issues, further straining the healthcare system's capacity to respond to the needs of the population, particularly vulnerable groups such as children[9].

The impact of war on children's health is multifaceted and varies depending on the conflict's nature and the country's development level[13]. Conflict also disrupts health reporting systems, making it difficult to obtain accurate assessments, since they often are influenced by biased sources[14]. Despite documentation of the Russo-Ukrainian War's general effects on healthcare, the specific impact on children's health in this region remains underexplored in the scientific literature[14]. This gap in research is alarming, given the conflict's potential long-term consequences on a generation of children. Understanding how modern conflict affects paediatric healthcare services is crucial for providing appropriate support.

This review is also framed against the backdrop of the July 8, 2024, missile attack on the Okhmatdyt Children's Hospital in Kyiv, the biggest children's hospital in the country[15]. Beyond the immediate casualties, the destruction of critical healthcare infrastructure emphasises the long-term impact on care provision and the urgency of this research. The review seeks to compile available open-source information on how the first two years of the Russo-Ukrainian War have impacted child health in Ukraine. In doing so, it aims to uncover paediatric health challenges and provide a foundation for future research efforts.

## Methods

Given the complex nature of the Russo-Ukrainian War and its impact on paediatric health, a systematic scoping review methodology was selected, guided by the frameworks of Arksey and O'Malley[16] and Levac et al.[17]. This approach allows for a comprehensive examination of the available literature to identify key concepts, gaps, and evidence within this emerging field, as well as to integrate and synthesise findings from various types of studies.

A scoping review is especially valuable when a particular research area remains underexplored or when the existing data are highly complex or heterogeneous. In this context, the variability and quality of the data are compounded by the ongoing conflict and limit the applicability of a traditional systematic review. Scoping reviews offer a broader examination of the literature, enabling researchers to map out key concepts, identify gaps, and gain an overview of the available evidence. This approach is particularly well-suited for emerging fields where the evidence base is not yet fully developed, allowing for a more flexible and comprehensive synthesis of diverse sources.

Adopting an interdisciplinary approach, this scoping review combines medical and social science perspectives to identify research gaps. The review aims not only to document the disruption of paediatric healthcare services due to the Russo-Ukrainian War but also to frame this understanding within a global context, and to highlight critical areas for further investigation. This methodology not only broadens the scope of the review but also enhances the potential to develop targeted interventions and policies to better support affected children.

The methodology was guided by a predetermined protocol in alignment with the PRISMA Extension for Scoping Reviews (PRISMA-ScR).

### Search strategy

The foundation of this review is based on addressing the following research questions: What has been reported regarding the impact of the Russo-Ukrainian War on children's health in Ukraine? What relevant studies can be identified? What study designs have been employed?

To capture relevant studies, a targeted search was conducted on PubMed on September 7, 2023. Since the effects of war on health are multifaceted, the search strategy was designed to be as broad as possible to encompass different aspects. The search terms included ""Ukraine OR Ukrainian" AND "child OR children"". The search was confined to articles published after the full-scale invasion on February 24, 2022, to try to target articles referring to potential effects of the war. The broad search enabled the scoping review to encompass many potential aspects and effects of the war. All articles from the search were initially retrieved, totalling 559 articles.

### Screening process

The primary screening process was facilitated using the "Rayyan" screening tool. Three team members independently reviewed the titles and abstracts of the retrieved articles. Any article that received two votes for "preliminary eligible" as per the inclusion criteria listed below was included for further full-text screening. Discrepancies in decisions were resolved through consensus meetings among the reviewers, ensuring that different perspectives were considered and potential biases were minimised.

**Inclusion and exclusion criteria.** Articles were assessed based on pre-defined *inclusion criteria*: relevance to the paediatric population in Ukraine, specific to the context of the Russo-Ukrainian war and focus on health-related impacts.

All articles deemed possibly eligible underwent a full-text review. Articles were excluded if inaccessible during the review period. Additionally, articles were excluded if it was revealed at this stage that they were not relevant to the geographical location, time, or content focus (e.g., those addressing issues outside of Ukraine or not specific to children's health impacts during the conflict). Articles available only in languages not accessible to the research team were also excluded.

### Data extraction and synthesis

Following the screening process, data extraction was performed on the eligible articles. The extraction process involved summarising key information from each article, including population, health outcomes, and the context of the findings. Initial head topics were identified based on common themes that emerged during data extraction. The full list of themes includes Healthcare System, Injury and Trauma Care, Perinatal and Maternal Health, Chronic Diseases, Cancer, Institutionalised Children, Disabilities, Mental Health, and Refugee Health

To ensure a comprehensive synthesis, the research team organised the articles based on the identified themes and conducted a second survey of the articles. The reviewers of the research team were allocated specific topics and were responsible for analysing and presenting the findings within their assigned areas. This process allowed for a refined thematic analysis, where patterns and insights were systematically categorised into the identified themes aligned with the research questions.

Subsequently, a thematic analysis of the scoping review findings was conducted to systematically categorise the data into relevant themes aligned with the research questions. This process allowed for identifying and organising key patterns within the literature, allowing for a deeper understanding of the topic under investigation.

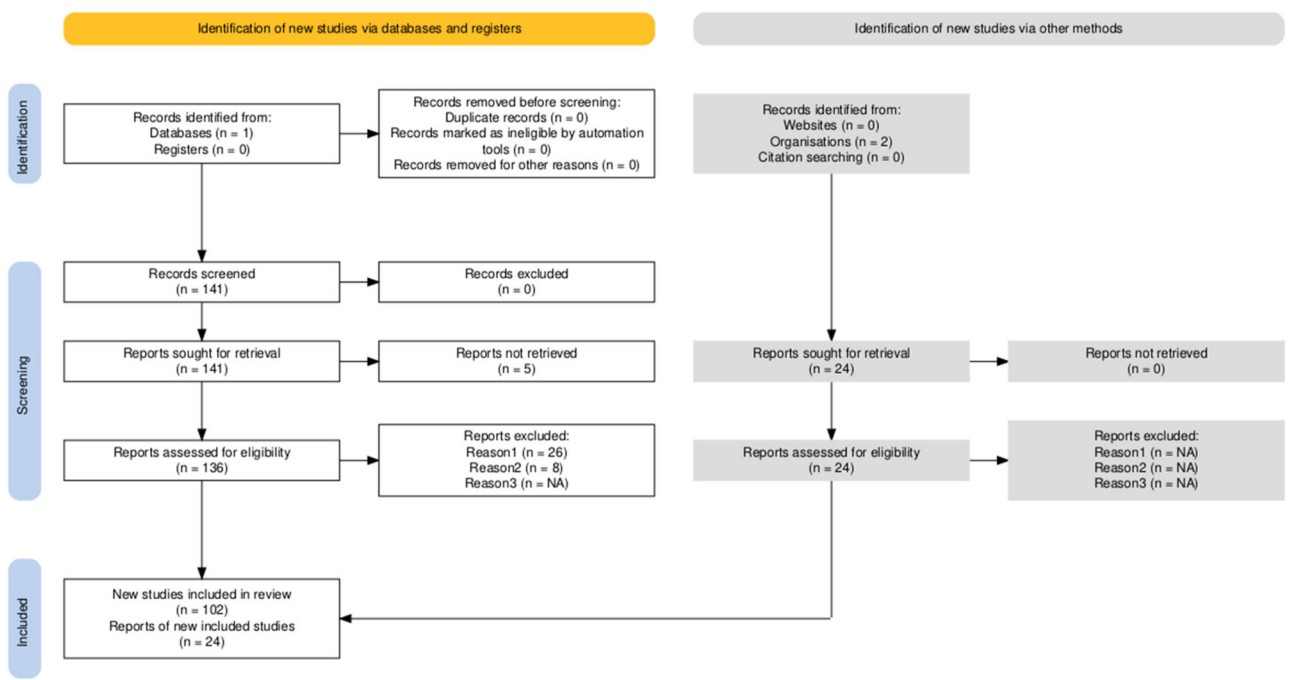

**Fig. 1 | Flow diagram of the inclusion process.** The diagram shows a visual representation of the decision-making and refinement steps involved in the inclusion process.

## Cross-reference with grey literature

Given the limited availability of empirical data directly related to the health effects on children in Ukraine, the review was supplemented with situational reports from the World Health Organisation (WHO) and UNICEF. These reports provided additional context and data to support the claims made in the reviewed articles. The inclusion of grey literature helped bridge gaps where peer-reviewed studies were lacking, especially in ongoing conflict settings where data collection is challenging. The supplementary data were selected based on relevance to the geographical location (Ukraine) and time period (February 24, 2022, to September 7, 2023).

## Flow diagram

The scoping review process, as shown in the flow diagram, systematically identified, screened, and included relevant studies. Initially, 559 records were found during the targeted search on PubMed. After screening, 418 records were excluded based on predefined criteria, leaving 141 reports for retrieval. Of these, 136 were retrieved and assessed for eligibility, resulting in 26 exclusions due to irrelevant geography, time, content, or language barriers. Further exclusions during a second survey focused on studies not directly addressing children's health in Ukraine within the context of the Russo-Ukrainian War. Ultimately, 102 studies were included, along with 24 additional reports from organisational sources, totalling 126 studies. This process ensured that only relevant studies were included, providing a solid foundation for analysing the impact of the Russo-Ukrainian War on paediatric healthcare in Ukraine. The process is visually represented in Fig. 1, which was created using a PRISMA Flow Diagram Generator[18].

**Ethics approval and consent to participate.** This study was performed in accordance with the ethical standards as laid down in the 1964 Declaration of Helsinki and its later amendments. Ethical approval was not necessary for this study, as it involved a literature review rather than empirical research involving human subjects or the handling of sensitive personal data. This decision is in line with the regulations set forth by the Swedish Research Council (2017).

**Consent for publication.** Not applicable. This scoping review was not pre-registered.

## Results
### Overview of included studies

The scoping review included a comprehensive analysis of 102 articles, covering a broad spectrum of topics related to the impact of the Russo-Ukrainian War on paediatric health. The studies were sourced from PubMed and included a mix of letters, original research, reviews, editorials, commentaries, and case reports. The articles were published between February 24, 2022, and September 7, 2023, reflecting the immediate and ongoing effects of the conflict on child health in Ukraine. The final articles that were used in this article are listed together with the identified type of article, main and subtopics in Supplementary Data.

**Types of studies.** The most common type of article was letters, accounting for approximately 30% of the total articles included. These were primarily focused on the immediate challenges faced by healthcare providers and the broader public health implications of the conflict. Original research articles constituted about 25% of the studies, providing empirical data on specific health outcomes such as mental health impacts and the spread of infectious diseases. Reviews made up about 10% of the total, offering synthesised insights into broader topics such as chronic disease management and refugee health. The remaining articles included commentaries, case reports, and joint statements, which provided expert opinions, case analyses, and collaborative insights from international health bodies. The distribution of types of articles is visually represented in Fig. 2.

**Geographical and thematic focus.** The articles predominantly focused on the impact of the war within Ukraine, particularly in regions most affected by the conflict, such as the eastern territories. However, there was also significant attention given to the health challenges faced by Ukrainian refugees in host countries, including neighbouring nations like Poland, Germany, and Romania.

Thematically, the articles covered a wide range of topics, and the distribution of the articles' topics is visually represented in Fig. 3.

**Fig. 2 | Distribution of types of articles.** The figure showcases the distribution of types of articles identified during the scoping review.

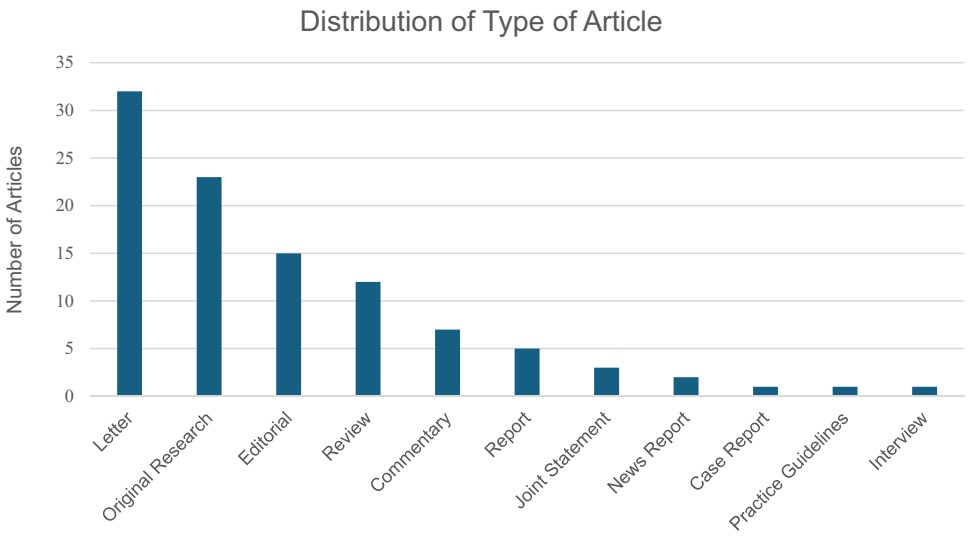

**Fig. 3 | Distribution of article topics.** The figure showcases the distribution of article topics, including both main and subtopics, identified during the scoping review.

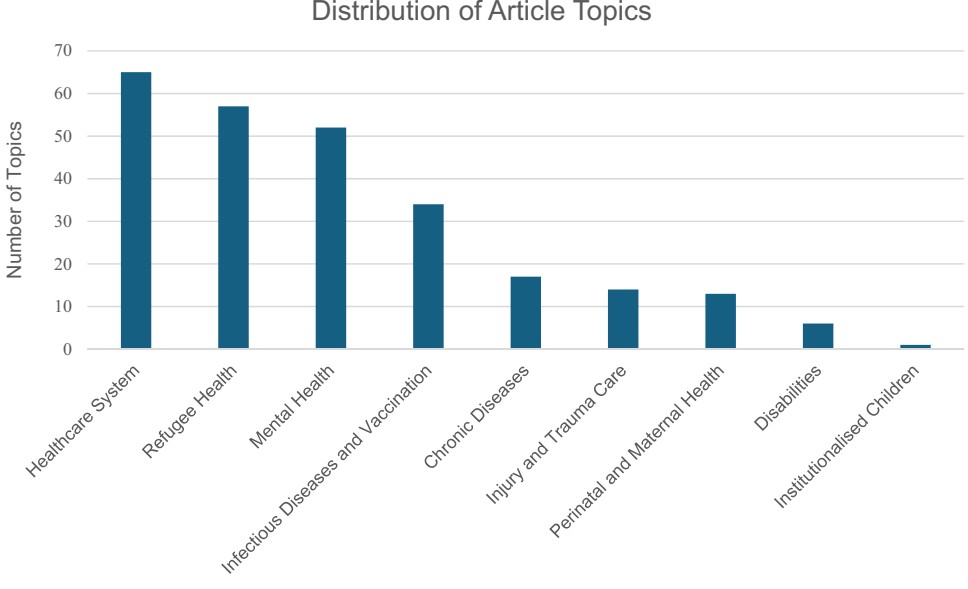

**Key findings.** The review identified a consistent theme of severe disruption in healthcare access, with many studies documenting the impact of the conflict on both physical and mental health services for children. The included studies provided a multi-layered view of the toll that the Russo-Ukrainian War has taken on paediatric health, while also identifying key areas where further research and policy intervention are needed. The following sections provide a detailed breakdown of the findings, highlighting both the immediate and long-term impacts on paediatric healthcare in Ukraine.

### Disrupted access to essential and healthcare services

The invasion of Ukraine in February of 2022 and subsequent warfare have impacted access to basic services essential for survival and to the healthcare system at large[19–82]. Disruptions in electricity have hindered food storage and heating, leading to increased food prices and compromised living conditions[19,26,28,29]. In the Mariupol Oncological Dispensary, generators intended for upholding advanced treatments were being used sparingly to prioritise cooking and feeding patients and staff over medical uses[31]. However, this situation was not representative of all affected regions in Ukraine.

The Russian invasion has severely disrupted access to healthcare services across Ukraine[22,29,83–87]. Emergency medical services face blockades and shelling, hindering the transfer of critically ill patients, including children[22,29]. Russian forces have denied ambulances access to occupied cities, further exacerbating the crisis[22].

Primary healthcare services such as vaccinations and routine checkups have largely been suspended due to a reallocation of resources towards treating war-related injuries[31,88]. Despite these challenges, a pilot programme has been initiated to expand the newborn genetic screening programme from covering four to 21 disorders, including conditions such as spinal muscular atrophy[80].

The war has also affected healthcare workers[19,20,32,33,63,72,77,78]. The invasion has led to a shortage of doctors, with many forced to relocate due to safety concerns[19]. Healthcare workers themselves experience heightened stress and anxiety, further impacting the workforce[19]. Women make up 83% of the country's health workforce but also compose the majority of migrants and internally displaced people[21,37]. Medical education has also been disrupted, with students leaving the country or joining the military[24]. In occupied regions, the departure of essential personnel like oncologists and radiation safety officers has compromised equipment control and safety[31].

Disruptions in traditional communication channels due to power outages and mobile limitations have prompted doctors to adopt telemedicine and to use messaging platforms for patient communication[19,28,31]. However, critics warn that virtual healthcare systems make them increasingly more vulnerable to cyberattacks[29].

## Impact on specific health issues

At the heart of this conflict's impact on paediatric health are specific challenges that exacerbate an already dire situation. These include difficulties in managing direct injuries, chronic diseases, disrupted maternal and perinatal health services, and an increase in infectious diseases due to vaccine programme interruptions. A breakdown of these key health issues is provided below.

**Injury and trauma care.** Children have repeatedly been exposed to what appears to be deliberate bombings of schools, hospitals, and playgrounds[36,89]. Reports also indicate illegal deportation and hostage-taking of children from Ukraine to Russia or Russian-held territories[36,89].

The war has inflicted direct injuries on children through explosive weapons and gunfire[22,25,28,34–36,70,81,82,89–93]. Children suffer severe burns, head and neck trauma, and penetrating injuries, which require specialised medical care and disproportionate levels of health services[81,91].

The direct effects of the war on children's health are also illustrated by case reports such as that of a 16-year-old boy who was shot at close range to the back of the neck and treated at National Pirogov Memorial Medical University, Vinnytsya, Ukraine[34]. The bullet injured his spinal cord, resulting in left upper limb paralysis and loss of sensation.

Gunshot trauma in children requires immediate evaluation and surgical intervention to improve postoperative recovery and physical therapy outcomes. Optimal interventions require access to ultrasound, computed tomography, and magnetic resonance imaging, alongside exploratory surgery[34]. All gunshot and shrapnel wounds are also considered conditionally contaminated and require broad-spectrum antibiotic therapy.

Traumatic experiences may lead to both physical and long-term psychological trauma, with children suffering from PTSD, anxiety, depression, and other mental health issues[21,28,45,60,63,64,89–91,93–95]. There is little documentation of physical rehabilitation needs that the war produces apart from general guidelines emphasising that rehabilitation should be active, long and continuous[34].

**Perinatal and maternal health.** The war has substantially affected perinatal and maternal health[20,21,23,30,37–40,79,80,96–98]. Pregnant women face limited access to maternal healthcare due to infrastructure damage and restricted medical treatment availability[40,85,88,99], which increases the risk of obstetric complications as well as maternal and neonatal mortality[21,40,83,88]. Reports from the World Health Organisation underline the limited data pertaining to maternal and newborn health[100].

Perinatal centres in Ukraine have operated under extreme conditions, with deliveries and caesarean sections performed in bomb shelters during airstrikes[37,100]. Several hospitals have been repurposed from providing essential services and primary healthcare to supporting and providing care for conflict-related trauma, leading to disruptions in basic and routine healthcare services, including maternal and child health[88]. This shift in priorities reflects the overwhelming demands placed on healthcare systems during the conflict, compromising routine but essential care services. Higher prevalence of preterm babies has been noted in conflict zones[60,100]. This is particularly concerning, considering specialised care required for premature and at-risk infants is likely unavailable in these regions[100]. Negative experiences among pregnant women also stem from fear, uncertainty, and lack of support during the war[38].

Millions of Ukrainian children under five and pregnant/breastfeeding mothers urgently need nutritional aid, especially in war-torn eastern regions[101]. Distributing breast milk substitutes is crucial since breastfeeding rates are low[88,101,102]. Breast milk has sometimes been the only available source of nutrition in shelters[37]. Certain perinatal centres have managed to secure breast milk for critically ill children through limited donor programmes[37]. Internet resources have been established to provide information on breastfeeding, and new guidelines have been published regarding infant and young child feeding following a nuclear emergency[37].

The war has also shed light on the complexities of surrogacy in Ukraine, where commercial surrogacy has been legal since 2002[39,79]. The armed conflict has exposed vulnerabilities within the system, leaving children born through surrogacy stranded and surrogate mothers in Ukraine facing legal issues[39].

**Chronic diseases.** It is well documented that children with chronic conditions have been impacted by this war[24,27,30,33,42,67,68,70,73,80,92,98,103–107]. Children with chronic illnesses face difficulties managing their conditions due to limited access to medication and essential equipment[19,107–109]. The war has led to fewer patient visits and reduced availability of care[19]. Additionally, conflict may have led to preventable complications from delayed diagnosis and treatment[19,83].

**Cancer.** Wars consequence on cancer care is also highly apparent[24,30,31,43,44,68,73,74,108,110,111]. Cancer poses a significant challenge, with hundreds of children at risk of premature death due to treatment interruptions, stress, and increased infection risk[112]. Cancer facilities have closed or reduced the number of treatments provided due to power shortages, forcing centres in western Ukraine to accommodate twice the normal patient-load[31,113]. This increase in patient load in western Ukraine leaves hospitals vulnerable, with little capacity for handling any additional influx of patients or unforeseen crises[112].

**Institutionalised children with disabilities.** Prior to the conflict, Ukraine had one of the highest rates of institutionalised children, over half with disabilities[58]. Children with disabilities have been affected by the invasion[24,25,29,58,64,68] and represent a group particularly at risk of exclusion from social protection and neglect, abandonment and death[24,55]. It is imperative for social care institutions and public health to ensure that these individuals are recognised and have access to professional care[24].

**Infectious diseases and vaccination challenges.** The war has impacted the spread of infectious diseases and contributed to even lower vaccination rates[24,29,30,41,44–53,57,67,68,71,76,90,92,96–98,105,114–122]. Ukraine already had a high prevalence of infectious diseases such as HIV and Tuberculosis prior to the invasion in 2022, compared to other European countries[46,114]. The war has worsened access to diagnostics and treatment, while crowded conditions in displacement settings increase the risk of spreading these diseases[46,83,114,123].

Ukraine's vaccination rates are among the lowest in Europe, with vaccination efforts hampered as a consequence of war[52,88,92,113,114]. Vaccine hesitancy remains high, especially in rural areas, contributing to the challenges of maintaining immunisation programmes during conflict[76,92]. Disrupted national vaccination programmes have led to low immunisation rates for diseases such as polio, measles, and COVID-19[52,88,92,96,98,113,114,122–124], posing significant risks not only within Ukraine but also in host countries due to the potential of outbreaks of vaccine-preventable diseases among refugee populations[124].

The rise of antimicrobial resistance is a particular fear, with multi-resistant bacteria posing a threat to both children and those with war injuries[103,119]. The problem is exacerbated by the overuse of broad-spectrum antibiotics in conflict zones where hygiene is poor and deep, contaminated wounds are common[125–127]. Laboratory assessments in Ukraine identified multiple challenges, including inadequate quantities of automated microbiology equipment, suboptimal laboratory quality, and limitations in information management systems[127]. The mass exodus of people raises concerns about the potential spread of resistant bacterial strains, highlighting the need for strict antibiotic surveillance[46,114,127].

**Mental health.** The acute stress of the conflict has exacerbated mental health effects of the COVID-19 pandemic, straining an already resource-limited system[27,32,55,58,83,124,128–130]. Many articles warn about the inevitable effects of the war on paediatric mental health[23,25–27,29,30,32,35,37,38,43,54–64,66–68,72,75,77,78,82,89–96,98,103,121,128,129,131–139].

While research on Ukrainian children's mental health is partial, surveys reveal significant issues among refugees, including self-mutilation, intoxications, and insomnia[58,92]. A study estimated that 20% of Ukrainian children have endured at least two adverse life experiences, such as physical or sexual abuse, domestic violence, or household issues such as substance abuse and criminality[96,132].

Specialists warn that untreated psychological trauma can have long-lasting effects, potentially impacting future generations, and there is an urgent need for active screening and trauma-focused interventions[55,60,77,93,132,134,138,140]. However, efforts are hindered by a lack of trained specialists within paediatric psychiatry and appropriate assessment tools for younger children[56,63,78,131,136,139].

## Challenges for displaced children and receiving countries

The invasion has resulted in significant internal and external displacement of people[21,24–27,41,45,46,51–54,57,59,61,63–69,71,73,74,89,92,94–98,103,105,107,108,110,111,114–117,119–122,128,129,131–138]. The displacement has strained evacuation centres with overcrowding, inadequate heating, and increased risk of communicable diseases[21,26,83,120,121]. Not only are paediatricians limited in these settings[24], concerns about sexual and gender-based violence have emerged within these displacement settings as well[67,83,101,124].

Neighbouring countries face significant challenges in providing healthcare for the sudden influx of refugees, mainly consisting of women and children[67]. International organisations and NGOs have mobilised additional staff to address these needs[66,67]. However, challenges persist, including coordination issues, inadequate healthcare system readiness, and limited cross-border collaboration[67]. Action is necessary to adapt guidelines, establish safe reception centres, improve migration governance, and provide essential tools and training for healthcare workers to meet evolving needs[67]. There is a proposal for coordinated military-civilian transfer of severe paediatric war zone trauma victims to neighbouring countries to improve patient outcomes[92].

Host countries face logistical challenges in organising assessments and treatment plans for disabled and cancer-diagnosed children[65,68]. Special attention must be given to children with complex needs to ensure their access to specialist healthcare during resettlement programmes[92]. Healthcare professionals experience a lack of guidelines, language barriers, and limited medical history upon refugees' arrival[26,65,92]. Converting prescriptions to national equivalents and addressing differences in immunisation schedules pose additional challenges[105]. Navigating foreign healthcare systems, compounded by language barriers, further impedes access to care[61]. Addressing these barriers is essential for the well-being of Ukrainian refugees in host nations.

## Discussion

The most critical finding from this scoping review is the far-reaching impact of the war on paediatric healthcare in Ukraine, exposing a deeply disrupted system in which both direct and indirect consequences significantly affect children's health. These effects range from acute physical injuries to long-term challenges, such as reduced access to routine medical care. However, a major barrier to fully understanding and addressing these impacts is the lack of robust empirical data, which limits reliable and comprehensive analyses.

An important finding from this study is the surprisingly limited amount of literature that specifically addresses the situation of children in Ukraine. This discrepancy raises the critical question of why the plight of children, despite their vulnerability during the ongoing conflict, has not received more focused attention in the scientific community. The scarcity of evidence-based articles during the initial phase of the conflict could potentially be attributed to factors such as the prioritisation of immediate crisis response over research efforts and concerns about publishing data that could potentially benefit the adversary. The fact that "Refugee Health" was seen as one of the major themes could be due to the fact that they, to a larger degree, are written and published by host countries.

Many articles focus on the potential risks of war in general, often extrapolating findings from previous conflicts to the Russo-Ukrainian War. Acts of resilience and adaptation by healthcare providers are noted, although these are often described in broad, generic terms. There is a critical need for more detailed documentation of these efforts to better appreciate and learn from the specific strategies employed to sustain paediatric healthcare amid conflict. This deeper insight is essential for developing targeted interventions and guiding future research to support healthcare systems in war-torn regions. Specific case studies or detailed accounts of successful interventions could serve as valuable resources for healthcare providers in similar settings.

The findings of this review align with paediatric healthcare outcomes reported from other modern armed conflicts, with disrupted routine medical services, inadequate prenatal and chronic illness care, and challenges in addressing mental health needs[141–149]. A recent scoping review, published after this review was conducted, highlights the broader mental health burden of war, further emphasising gaps in empirical studies and unmet needs across populations, including children[150]. In the absence of substantial empirical data from Ukraine, prior modern conflicts were revisited to identify systematic gaps in research related to paediatric health in conflict settings. While historical data provides valuable context, it is important to recognise that the unique characteristics of the current conflict, such as its scale, the involvement of urban centres, and the use of modern warfare technologies, may result in different healthcare needs and outcomes.

Due to the high volume of trauma and trauma surgeries, previous conflicts have documented delayed treatment for children with routine surgical conditions like cleft lip and palate, indicating secondary suffering due to the focus on war-related injuries[142]. Research addressing the prevalence and impact of delayed treatment, resource allocation, and management is required to better understand the full-scale impact of the war. Additionally, future research should explore the long-term consequences of such delays on child development and quality of life, which are critical yet often overlooked aspects of paediatric healthcare in conflict zones.

The trauma from war necessitates extensive physical therapy and rehabilitation services[151]. However, insufficient research and data on injury types and numbers often make it difficult to comprehend the true scale of rehabilitation needs[151]. Previous conflicts have shown a mismatch between rehabilitation needs and available resources, with significant delays in care delivery[152]. Additionally, research often neglects essential rehabilitation services and the specific needs of children with complex injuries, leading to concerns about the long-term effectiveness of interventions[153]. Future research should focus on detailed documentation and analysis of rehabilitation needs to enhance care delivery and improve outcomes. It is also important to consider the integration of community-based rehabilitation programmes, which may offer more sustainable and accessible options in the post-conflict recovery phase.

Historical data from modern conflicts also shed light on some of the long-term impacts of war on children's physical and mental health[154,155]. For instance, during the Balkan conflict, weapon-related deaths among children surged, and the psychological trauma, combined with easy access to weapons, created a hazardous environment that persisted long after the war[154,156]. Post-war environments are characterised by the presence of unexploded ordnance, posing ongoing dangers, particularly to children[142]. The detrimental effects of war will not be limited to active hostilities, and post-conflict response plans need to be prepared to tackle these challenges. Proactive planning for post-conflict recovery could include risk education for children and families about unexploded ordnance, as well as the establishment of dedicated mental health services to address ongoing trauma.

Armed conflict exacerbates mental health issues, and the long-term psychological effects can span generations. For instance, wartime evacuations during World War II resulted in higher rates of psychiatric disorders in adulthood for Finnish girls and subsequent mental health issues in their

**Table 1 | Listing the high-level topics and specific points of interest for future research initiatives generated by the scoping review**

| High-level topic | Points of interest |
|---|---|
| Healthcare infrastructure | Resilience and adaptability during conflicts[28,31] |
| | Impact of infrastructure damage on healthcare delivery[29,40] |
| | Strategies to safeguard and rebuild healthcare infrastructure[36,73] |
| Healthcare workers | Workforce availability and retention during conflict[32,78] |
| | Training and support mechanisms for healthcare workers in conflict and post-conflict settings[28,77] |
| Supply chain and logistics | Ensuring continuity of medical supply chains during conflict[29,35] |
| | Innovative logistics solutions to overcome supply chain disruptions in war zones[31,33] |
| Telemedicine and virtual healthcare | Effectiveness and challenges of implementing telemedicine in conflict areas[19,28] |
| | Security and privacy concerns associated with virtual healthcare systems in war zones[20,29] |
| Injury and trauma care | Best practices for managing acute war-related injuries in children[36,92] |
| | Long-term outcomes and rehabilitation strategies for paediatric trauma patients[34,36] |
| Rehabilitation needs | Comprehensive assessment of rehabilitation requirements for war-injured children[34,36] |
| | Development of specialised rehabilitation programmes and resources for paediatric patients[34,36] |
| Chronic diseases | Continuity of care for children with chronic illnesses during conflict[104,108] |
| | Impact of war on disease management and long-term health outcomes[31,33] |
| Infectious diseases and vaccination | Effects of conflict on the spread and treatment of infectious diseases[46,120] |
| | Strategies to maintain vaccination programmes and address vaccine hesitancy during war[92,122] |
| Antibiotic resistance | Monitoring and managing antibiotic resistance in conflict settings[71,119] |
| | Development of guidelines for the use of antibiotics in war-related injuries[71,119] |
| Mental health | Prevalence and treatment of mental health issues in children affected by war[55,128] |
| | Long-term psychological impacts of conflict on children and effective intervention strategies[30,60] |
| Perinatal and maternal health | Access to maternal and newborn healthcare services during conflict[21,40] |
| | Impact of war on maternal and neonatal outcomes and strategies to mitigate risks[20,30] |
| | Legal and ethical considerations of surrogacy[39,79] |
| Disability and special needs | Addressing the needs of children with disabilities in conflict zones[24,55] |
| | Ensuring accessibility and support for disabled children during and after the war[24,25] |
| Refugee health | Health challenges faced by internally displaced persons and refugees[96,105] |
| | Coordination and provision of healthcare services in refugee camps and host countries[57,92] |

offspring[157,158]. Despite various interventions during conflicts, significant gaps remain in understanding their efficacy and implementation[159–163]. This calls for longitudinal studies and targeted mental health interventions to address the enduring psychological impacts of war on children and their families. Future interventions should be informed by culturally sensitive approaches and should consider the unique psychological profiles of children who have been exposed to different types and levels of trauma.

Studies and systematic reviews published after this review continue to report similar problems and advocate for more empirical studies on the social and health outcomes for children[30,35,164–167]. Going forward, risk matrix approaches could be used to prioritise research efforts by assessing both the impact of the issue and the likelihood/frequency with which it occurs. For example, mental health could be categorised as high impact and high likelihood, given the severe and widespread consequences indicated in Ukraine and documented during previous conflicts. Utilising a risk matrix could also help in the strategic allocation of resources, ensuring that the most critical issues receive the attention and funding needed to reduce their impact.

Conducting research during ongoing conflict is challenging but essential[168,169]. High-quality research requires adequate funding and research competence, emphasising the need for collaborative efforts. Partnering with local universities, research institutions, and healthcare providers can enhance the research competence and ensure that the research is contextually relevant[170]. Capacity building through training local researchers and healthcare workers is also critical to ensure sustainability[171]. Given the ensuing war, ensuring the safety of both patients and researchers is

important, and remote data collection and implementation of robust security protocols need to be considered[168].

The review has identified several areas of research, ranging from the war's effect on healthcare infrastructure to the delivery of healthcare services. The findings of this review have been synthesised into high-level topics, with specific points of interest for future research identified. For a more detailed summary, consult Table 1 below. As the conflict evolves, continuous updates to this research agenda will be necessary to address emerging health challenges and adapt to the changing landscape of paediatric healthcare needs.

This scoping review acknowledges several biases and limitations. Firstly, the focus on articles published post-February 24, 2022, introduces selection bias, potentially overlooking valuable pre-conflict data. The exclusion of non-English articles, particularly those in Ukrainian or Russian, introduces language bias. The reliance on PubMed as the sole database may have led to publication bias, missing studies from non-indexed journals or grey literature. Expanding the search to include additional databases might have provided a broader perspective. The consensus meetings to resolve screening disagreements could have introduced confirmation bias. Furthermore, inconsistencies in data extraction due to the lack of a standardised protocol may affect the reliability of findings. Practical limitations, such as inaccessibility of certain articles and data source heterogeneity, posed challenges for data synthesis. Supplementing empirical data with WHO and UNICEF reports may introduce information bias. The review relies heavily on anecdotal evidence and open letters, complicating the quantification of conflict impact on paediatric health. Moreover, existing databases also often

aggregate conflict mortality data without isolating impacts on children, further restricting the robustness of the findings.

Despite its limitations, this review provides timely insights into the healthcare challenges faced by children in Ukraine during the current conflict, ensuring relevance to ongoing issues. The focused scope allowed for an in-depth examination of specific paediatric healthcare concerns, contributing valuable knowledge to a relatively underexplored area. By supplementing data with reputable sources such as WHO and UNICEF reports, the review enhances its credibility, offering an inclusive perspective on the impact of the conflict on children's health. Additionally, the use of a structured consensus process for screening helped ensure a thorough selection of articles.

## Conclusion

The war in Ukraine has profoundly impacted children's health through direct injuries, disrupted chronic care, and long-term psychological trauma. The overwhelmed healthcare system and damaged infrastructure further complicate care provision, presenting significant barriers to addressing these challenges effectively. The findings of this scoping review highlight the multifaceted impacts of modern armed conflicts on paediatric healthcare and constitute yet another reminder of the poor adherence to the Geneva Conventions of 1949 that were adopted, among other things, to protect children and the healthcare system in wars[172]. The urgent need for more empirical data is clear, as a fuller understanding of these impacts is essential for informing effective interventions.

Future research should focus on detailed documentation of healthcare providers' resilience and adaptation strategies, comprehensive analysis of rehabilitation needs, longitudinal studies on the psychological impacts of war, and the development of tailored preparedness and response strategies. Specific areas for further exploration include in-depth case studies and comparative analyses of different regions and healthcare systems to understand effective resilience and adaptation strategies. This involves examining how healthcare workers manage resource scarcity, maintain service delivery under duress, and implement innovative solutions to emerging challenges.

A comprehensive analysis of the rehabilitation needs of children injured in conflict should identify physical, psychological, and social aspects, developing standardised assessment tools and protocols for various types of injuries and trauma. Longitudinal studies are necessary to understand the full spectrum of psychological effects on children exposed to conflict, tracking mental health outcomes over time, identifying factors that contribute to resilience, and evaluating the effectiveness of different therapeutic interventions.

## Data availability

The data supporting the findings of this study are included within the article. The sources of data used for this scoping review are publicly available. Additional details are available upon reasonable request and may be directed to the corresponding author.

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

## Acknowledgements
We would like to acknowledge the valuable input from healthcare professionals and researchers who provided insights that shaped the thematic analysis of this review. The research team would also like to extend their sincere thanks to Arita Holmberg, Associate Professor, Senior Lecturer in Political Science at Swedish Defence University, for reading and providing valuable comments on the text, which contributed to its improvement. No funding was received for this study.

## Author contributions
F.S. conducted the review of articles, synthesised the data, and wrote the final draft of the manuscript. S.F. conducted the review of articles, synthesised the data, and created the visual representations. S.P. conducted the review of articles, synthesised the data, and managed the references. A.W. provided support and contributed to the editing and revision of the final draft. A.A. supported the development of the methodology and overall study design. D.B., as the scientific supervisor, provided support and contributed to the editing and revision of the final draft. All authors have read and agreed to the published version of the manuscript.

## Funding

## Competing interests
The authors declare no competing interests.
