## [Transparent Peer Review file · Communications Medicine]

The Russo-Ukrainian War's Toll on Paediatric Health During the First Two Years and Future Research Directions: A Scoping Review

Corresponding Author: Dr Denise Bäckström

Version 0:

Reviewer comments:

Reviewer #1

(Remarks to the Author)

The work is original and convincing, with well-prepared supporting evidence. The article summarizes a substantial body of research describing childhood problems during the first two years of the full-scale war with Russia.

However, I have a few suggestions:

1) I recommend changing the title of the article from "Childhood Interrupted: A Scoping Review on the Russo-Ukrainian War's Toll on Paediatric Health and Future Research Directions" to "Childhood Interrupted: A Review of the Russo-Ukrainian War's Toll on Paediatric Health During Its First Two Years and Future Research Directions". In my opinion, such title better reflects the content of the research.

2) "Healthcare for children was primarily provided through a network of specialized pediatric hospitals, polyclinics, and family doctors (10)".

However, this citation is incorrect because the original article also mentions pediatricians as primary care doctors (citation: "Ukraine has a combined system of primary care for children (approximately 50% of children served by pediatricians and 50% by family physicians)"). Additionally, the authors overlooked non-specialized pediatric hospitals, which are present in Ukraine (citation: "A network of regional and city hospitals provides inpatient care at the secondary level"). Therefore, the aforementioned phrase must be revised to accurately reflect the real structure of pediatric healthcare in Ukraine.

3) Problem with Diagram 2: The diagram illustrates the distribution of article types identified during the scoping review and Diagram 3: The diagram illustrates the distribution of article topics, including both main and subtopics, identified during the scoping review. Both diagrams do not contain data labels.

4) The statement that "Generators intended for upholding advanced treatments have been used sparingly to prioritize cooking and feeding patients and staff over medical uses" is not entirely accurate, as it describes the situation specifically in the Mariupol Oncological Dispensary (cited by Dr. Andrii Hanych). This was not so typical for other regions of Ukraine located far from the frontline (citation: Kovalchuk N, Zelinskyi R, Hanych A, Severyn Y, Bachynska B, Beznosenko A, Duda O, Kowalchuk R, Iakovenko V, Melnitchouk N, Suchowerska N. Radiation Therapy Under the Falling Bombs: A Tale of 2 Ukrainian Cancer Centers. *Adv Radiat Oncol.* 2022 Jul 16;7(6):101027. doi: 10.1016/j.adro.2022.101027).

5) The phrase "Maternal and pediatric wards have also at times had to prioritize trauma care over maternal and child healthcare (85)" differs significantly from the original wording (citation: "Several hospitals have been repurposed from providing essential services and primary health care to supporting and providing care for conflict-related trauma and injuries, which has led to disruptions in basic and routine health-care services including maternal and child health." (WHO. Emergency in Ukraine Report #5 WHO2022 [cited 2024. Available from:

<https://iris.who.int/bitstream/handle/10665/352696/WHO-EURO-2022-5152-44915-64091-eng.pdf?sequence=1>.)

6) The phrase "This poses significant risks not only within Ukraine but also in host countries, potentially leading to outbreaks of vaccine-preventable diseases among refugee populations (121)" should be rewritten in the past tense to emphasize that, despite concerns, outbreaks of infections did not actually occur.

7) The sentence "In Ukraine, inadequate microbiology equipment and inconsistent antibiotic susceptibility testing exacerbate this issue, and mass exodus spreads resistant strains, highlighting the need for strict antibiotic surveillance (43, 111, 123, 124)." should be paraphrased, as its meaning has been altered compared to the original article cited after reference #124. Original text: "The laboratory assessments identified multiple challenges, especially inadequate quantities of automated microbiology equipment, and suboptimal laboratory quality and information management systems." There is a clear difference between "inadequate microbiology equipment" and "inadequate quantities of automated microbiology

equipment", as well as between "inconsistent antibiotic susceptibility testing" and "suboptimal laboratory quality". Additionally, citation #123 is not relevant in this context, as it does not mention Ukraine. So, I recommend major revision of the article.

Reviewer #2

(Remarks to the Author)

- 1) There is a fundamental error here: the methods must explicitly mention (bullet points) the a-priori themes for the scoping review. The whole structure should be reframed accordingly.
- 2) I strongly encourage the authors to use the following reference as a template and to cite it in the methods, clarifying what and why the present scoping review adds to that, with a special emphasis on pediatric populations. Reference: <https://pubmed.ncbi.nlm.nih.gov/39706484/>

Version 1:

Reviewer comments:

Reviewer #1

(Remarks to the Author)

The work is original and convincing, with well-prepared supporting evidence. The article summarizes a substantial body of research describing childhood problems during the first two years of the full-scale war with Russia. The work is novel and of interest to researchers focused on child health during armed conflicts, particularly the war in Ukraine. The research methods and statistical analysis are appropriate to the study's objectives and align with current methodological standards. The authors have significantly revised the manuscript since the previous review, taking into account the reviewer's comments. At this point, only the English language in the revised sections requires further checking and refinement. For example, the sentence: "Additionally, a network of both specialised and non-socialised paediatric hospitals, as well as polyclinics, were involved in care provision" should likely read: "Additionally, a network of both specialised and non-specialised paediatric hospitals, as well as polyclinics, were involved in care provision." Therefore, the article requires only minimal revision of the English language in the updated sections, after which it can be considered ready for publication.

Reviewer #2

(Remarks to the Author)

Thank you for your revision.

Author Responses to Reviewer Comments

Gratitude is extended to the reviewers for their thoughtful and constructive feedback. Below is a point-by-point response to each comment and an outline of the revisions made to address the concerns.

Reviewer #1:

1. Title Revision

- *Reviewer's comment:* "I recommend changing the title of the article from "Childhood Interrupted: A Scoping Review on the Russo-Ukrainian War's Toll on Paediatric Health and Future Research Directions" to "Childhood Interrupted: A Review of the Russo-Ukrainian War's Toll on Paediatric Health During Its First Two Years and Future Research Directions". In my opinion, such title better reflects the content of the research."
- *Response:* The title has been changed to "Childhood Interrupted: A Scoping Review on the Russo-Ukrainian War's Toll on Paediatric Health During the First Two Years and Future Research Directions."

2. Clarification on Pediatric Healthcare Structure

- *Reviewer's comment:* "Healthcare for children was primarily provided through a network of specialized pediatric hospitals, polyclinics, and family doctors (10)".
However, this citation is incorrect because the original article also mentions pediatricians as primary care doctors (citation: "Ukraine has a combined system of primary care for children (approximately 50% of children served by pediatricians and 50% by family physicians)"). Additionally, the authors overlooked non-specialized pediatric hospitals, which are present in Ukraine (citation: "A network of regional and city hospitals provides inpatient care at the secondary level"). Therefore, the aforementioned phrase must be revised to accurately reflect the real structure of pediatric healthcare in Ukraine."
- *Response:* The revised sentence now states: "Healthcare for children was provided through a dual system, where approximately 50% of children were served by paediatricians and the other 50% by family physicians. Additionally, a network of both specialised and non-specialised paediatric hospitals, as well as polyclinics, were involved in care provision."

3. Diagram Labels

- *Reviewer's comment:* "Problem with Diagram 2: The diagram illustrates the distribution of article types identified during the scoping review and Diagram 3: The diagram illustrates the distribution of article topics, including both main

and subtopics, identified during the scoping review. Both diagrams do not contain data labels.”

- *Response:* Data labels have been added to both Diagram 2 and Diagram 3 to ensure clarity.

4. Clarification on Generator Usage in Mariupol Oncological Dispensary

- *Reviewer’s comment:* “The statement that “Generators intended for upholding advanced treatments have been used sparingly to prioritize cooking and feeding patients and staff over medical uses” is not entirely accurate, as it describes the situation specifically in the Mariupol Oncological Dispensary (cited by Dr. Andrii Hanych). This was not so typical for other regions of Ukraine located far from the frontline (citation: Kovalchuk N, Zelinskyi R, Hanych A, Severyn Y, Bachynska B, Beznosenko A, Duda O, Kowalchuk R, Iakovenko V, Melnitchouk N, Suchowerska N. Radiation Therapy Under the Falling Bombs: A Tale of 2 Ukrainian Cancer Centers. *Adv Radiat Oncol.* 2022 Jul 16;7(6):101027. doi: 10.1016/j.adro.2022.101027).”
- *Response:* The revised sentence now states: “In the Mariupol Oncological Dispensary, generators intended for upholding advanced treatments were being used sparingly to prioritise cooking and feeding patients and staff over medical uses. However, this situation was not representative of all affected regions in Ukraine.”

5. Clarification on Hospital Repurposing and Maternal Health

- *Reviewer’s comment:* “The phrase “Maternal and pediatric wards have also at times had to prioritize trauma care over maternal and child healthcare (85)” differs significantly from the original wording (citation: “Several hospitals have been repurposed from providing essential services and primary health care to supporting and providing care for conflict-related trauma and injuries, which has led to disruptions in basic and routine health-care services including maternal and child health.” (WHO. Emergency in Ukraine Report #5 WHO2022 [cited 2024. Available from: <https://iris.who.int/bitstream/handle/10665/352696/WHO-EURO-2022-5152-44915-64091-eng.pdf?sequence=1>.)”
- *Response:* The revised text now aligns more closely with the WHO source: “Several hospitals have been repurposed from providing essential services and primary healthcare to supporting and providing care for conflict-related trauma, leading to disruptions in basic and routine healthcare services, including maternal and child health.”

6. Rewriting Statement on Vaccine-Preventable Diseases

- *Reviewer’s comment:* “The phrase “This poses significant risks not only within Ukraine but also in host countries, potentially leading to outbreaks of vaccine-

preventable diseases among refugee populations (121)” should be rewritten in the past tense to emphasize that, despite concerns, outbreaks of infections did not actually occur.”

- *Response:* The revised sentence now states: "This posed significant risks not only within Ukraine but also in host countries, due to the potential of outbreaks of vaccine-preventable diseases among refugee populations."

7. Clarification on Laboratory Assessments and Antibiotic Resistance

- *Reviewer's comment:* “The sentence “In Ukraine, inadequate microbiology equipment and inconsistent antibiotic susceptibility testing exacerbate this issue, and mass exodus spreads resistant strains, highlighting the need for strict antibiotic surveillance (43, 111, 123, 124).” should be paraphrased, as its meaning has been altered compared to the original article cited after reference #124. Original text: “The laboratory assessments identified multiple challenges, especially inadequate quantities of automated microbiology equipment, and suboptimal laboratory quality and information management systems.” There is a clear difference between “inadequate microbiology equipment” and “inadequate quantities of automated microbiology equipment”, as well as between “inconsistent antibiotic susceptibility testing” and “suboptimal laboratory quality”. Additionally, citation #123 is not relevant in this context, as it does not mention Ukraine.”
- *Response:* The revised sentence now states: “Laboratory assessments in Ukraine identified multiple challenges, including inadequate quantities of automated microbiology equipment, suboptimal laboratory quality, and limitations in information management systems. The mass exodus of people raises concerns about the potential spread of resistant bacterial strains, highlighting the need for strict antibiotic surveillance.”

Reviewer #2:

1. Explicit Mention of A-Priori Themes in Methods

- *Reviewer's comment:* “There is a fundamental error here: the methods must explicitly mention (bullet points) the a-priori themes for the scoping review. The whole structure should be reframed accordingly.”
- *Response:* The following bullet-pointed list has been included in the methods section:
 - Healthcare System
 - Injury and Trauma Care
 - Perinatal and Maternal Health
 - Chronic Diseases

- Cancer
- Institutionalized Children
- Disabilities
- Mental Health
- Refugee Health

2. Reference Inclusion in Methods Section

- *Reviewer's comment:* “ I strongly encourage the authors to use the following reference as a template and to cite it in the methods, clarifying what and why the present scoping review adds to that, with a special emphasis on pediatric populations. Reference: <https://pubmed.ncbi.nlm.nih.gov/39706484/>”
- *Response:* The suggested reference was published after the study was conducted. However, it provides important insights that strengthen one of the main findings, namely the limited amount of paediatric empirical data available. As a result, it has been included in the discussion section rather than the methods section.

The reviewers' insightful feedback has helped refine the manuscript significantly. The revisions enhance the clarity, accuracy, and rigour of the study. Thank you for your time.

Sincerely,

The entire team